# The Last Iterate Advantage: Empirical Auditing and Principled Heuristic Analysis of Differentially Private SGD

## Abstract

We propose a simple heuristic privacy analysis of noisy clipped stochastic gradient descent (DP-SGD) in the setting where only the last iterate is released and the intermediate iterates remain hidden. Namely, our heuristic assumes a linear structure for the model.

We show experimentally that our heuristic is predictive of the outcome of privacy auditing applied to various training procedures. Thus it can be used prior to training as a rough estimate of the final privacy leakage. We also probe the limitations of our heuristic by providing some artificial counterexamples where it underestimates the privacy leakage.

The standard composition-based privacy analysis of DP-SGD effectively assumes that the adversary has access to all intermediate iterates, which is often unrealistic. However, this analysis remains the state of the art in practice. While our heuristic does not replace a rigorous privacy analysis, it illustrates the large gap between the best theoretical upper bounds and the privacy auditing lower bounds and sets a target for further work to improve the theoretical privacy analyses.

## 1 Introduction

Differential privacy (DP) [DMNS06] defines a measure of how much private information from the training data leaks through the output of an algorithm. The standard differentially private algorithm for deep learning is DP-SGD [BST14; ACGMMTZ16], which differs from ordinary stochastic gradient descent in two ways: the gradient of each example is clipped to bound its norm and then Gaussian noise is added at each iteration.

The standard privacy analysis of DP-SGD is based on composition [BST14; ACGMMTZ16; Mir17; Ste22; KJH20]. In particular, it applies to the setting where the privacy adversary has access to all intermediate iterates of the training procedure. In this setting, the analysis is known to be tight [NSTPC21; NHSBTJCT23]. However, in practice, potential adversaries rarely have access to the intermediate iterates of the training procedure, rather they only have access to the final model. Access to the final model can either be through queries to an API or via the raw model weights. The key question motivating our work is the following.

> Is it possible to obtain sharper privacy guarantees for DP-SGD when the adversary only has access to the final model, rather than all intermediate iterates?

### 1.1 Background & Related Work

The question above has been studied from two angles: Theoretical upper bounds, and privacy auditing lower bounds. Our goal is to shed light on this question from a third angle via principled heuristics.

A handful of theoretical analyses [FMTT18; CYS21; YS22; AT22; BSA24] have shown that asymptotically the privacy guarantee of the last iterate of DP-SGD can be far better than the standard composition-based analysis that applies to releasing all iterates. In particular, as the number of iterations increases, these analyses give a privacy guarantee that converges to a constant (depending on the loss function and the scale of the noise), whereas the standard composition-based analysis would give a privacy guarantee that increases forever. Unfortunately, these theoretical analyses are only applicable under strong assumptions on the loss function, such as (strong) convexity and smoothness. We lack an understanding of how well they reflect the "real" privacy leakage.

Privacy auditing [JUO20; DWWZK18; BGDCTV18; SNJ23; TTSSJC22; ZBWTSRPNK22] complements theoretical analysis by giving empirical lower bounds on the privacy leakage. Privacy auditing works by performing a membership inference attack [SSSS17; HSRDTMPSNC08; SOJH09; DSSUV15]. That is, it constructs neighbouring inputs and demonstrates that the corresponding output distributions can be distinguished well enough to imply a lower bound on the differential privacy parameters. In practice, the theoretical privacy analysis may give uncomfortably large values for the privacy leakage (e.g., $\varepsilon > 10$); in this case, privacy auditing may be used as evidence that the "real" privacy leakage is lower. There are settings where the theoretical analysis is matched by auditing, such as when all intermediate results are released [NSTPC21; NHSBTJCT23]. However, despite significant work on privacy auditing and membership inference [CCNSTT22; BTRKMW24; WBKBGGG23; LF20; SDSOJ19; ZLS23], a large gap remains between the theoretical upper bounds and the auditing lower bounds [AKOOMS23; NHSBTJCT23] when only the final parameters are released. This observed gap is the starting point for our work.

## 1.2 Our Contributions

We propose a *heuristic* privacy analysis of DP-SGD in the setting where only the final iterate is released. Our experiments demonstrate that this heuristic analysis consistently provides an upper bound on the privacy leakage measured by privacy auditing tools in realistic deep learning settings.

Our heuristic analysis corresponds to a worst-case theoretical analysis under the assumption that the loss functions are linear. This case is simple enough to allow for an exact privacy analysis whose parameters are can be computed numerically (Theorem 1). Our consideration of linear losses is built on the observation that current auditing techniques achieve the highest $\varepsilon$ values when the gradients of the canaries – that is, the examples that are included or excluded to test the privacy leakage – are fixed and independent from the gradients of the other examples. This is definitely the case for linear losses; the linear assumption thus allows us to capture the setting where current attacks are most effective. Linear loss functions are also known to be the worst case for the non-subsampled (i.e., full batch) case; see Appendix B. Assuming linearity is unnatural from an optimization perspective, as there is no minimizer. But, from a privacy perspective, we show that it captures the state of the art.

We also probe the limitations of our heuristic and give some artificial counterexamples where it underestimates empirical privacy leakage. One class of counterexamples exploits the presence of a regularizer. Roughly, the regularizer partially zeros out the noise that is added for privacy. However, the regularizer also partially zeros out the signal of the canary gradient. These two effects are almost balanced, which makes the counterexample very delicate. In a second class of counterexamples, the data is carefully engineered so that the final iterate effectively encodes the entire trajectory, in which case there is no difference between releasing the last iterate and all iterates.

**Implications:** Heuristics cannot replace rigorous theoretical analyses. However, our heuristic can serve as a target for future improvements to both privacy auditing as well as theoretical analysis. For privacy auditing, matching or exceeding our heuristic is a more reachable goal than matching the theoretical upper bounds, although our experimental results show that even this would require new attacks. When theoretical analyses fail to match our heuristic, we should identify why there is a gap, which builds intuition and could point towards further improvements.

Given that privacy auditing is computationally intensive and difficult to perform correctly [AZT24], we believe that our heuristic can also be valuable in practice. In particular, our heuristic can be used prior to training (e.g., during hyperparameter selection) to predict the outcome of privacy auditing when applied to the final model. (This is a similar use case to scaling laws.)

## 2 Linearized Heuristic Privacy Analysis

Theorem 1 presents our heuristic differential privacy analysis of DP-SGD (which we present in Algorithm 1 for completeness; note that we include a regularizer $r$ whose gradient is *not* clipped, because it does not depend on the private data $\mathbf{x}$). We consider Poisson subsampled minibatches and add/remove neighbours, as is standard in the differential privacy literature.

Our analysis takes the form of a conditional privacy guarantee. Namely, under the assumption that the loss and regularizer are linear, we obtain a fully rigorous differential privacy guarantee. The heuristic is to apply this guarantee to loss functions that are not linear (such as those that arise in deep learning applications). Our thesis is that, in most cases, the conclusion of the theorem is still a good approximation, even when the assumption does not hold.

Recall that a function $\ell : \mathbb{R}^d \to \mathbb{R}$ is linear if there exist $\alpha \in \mathbb{R}^d$ and $\beta \in \mathbb{R}$ such that $\ell(\mathbf{m}) = \langle \alpha, \mathbf{m} \rangle + \beta$ for all $\mathbf{m}$.

**Theorem 1** (Privacy of DP-SGD for linear losses). *Let* $\mathbf{x}, T, q, \eta, \sigma, \ell, r$ *be as in Algorithm 1. Assume* $r$ *and* $\ell(\cdot, x)$, *for every* $x \in \mathcal{X}$, *are linear.*

*Letting*

$$P := \mathsf{Binomial}(T, q) + \mathcal{N}(0, \sigma^2 T), Q := \mathcal{N}(0, \sigma^2 T), \tag{1}$$

*DP-SGD with* `last_iterate_only` *satisfies* $(\varepsilon, \delta)$-*differential privacy with* $\varepsilon \geq 0$ *arbitrary and*

$$\delta = \delta_{T,q,\sigma}(\varepsilon) := \max\{H_{e^\varepsilon}(P, Q), H_{e^\varepsilon}(Q, P)\}. \tag{2}$$

*Here,* $H_{e^\varepsilon}$ *denotes the* $e^\varepsilon$-*hockey-stick-divergence* $H_{e^\varepsilon}(P, Q) := \sup_S P(S) - e^\varepsilon Q(S)$.

---

**Algorithm 1** Noisy Clipped Stochastic Gradient Descent (DP-SGD) [BST14; ACGMMTZ16]

**function** DP-SGD($\mathbf{x} \in \mathcal{X}^n$, $T \in \mathbb{N}$, $q \in [0, 1]$, $\eta \in (0, \infty)$, $\sigma \in (0, \infty)$, $\ell : \mathbb{R}^d \times \mathcal{X} \to \mathbb{R}$, $r : \mathbb{R}^d \to \mathbb{R}$)

    Initialize model $\mathbf{m}_0 \in \mathbb{R}^d$.

    **for** $t = 1 \cdots T$ **do**

        Sample minibatch $B_t \subseteq [n]$ including each element independently with probability $q$.

        Compute gradients of the loss $\nabla_{\mathbf{m}_{t-1}}\ell(\mathbf{m}_{t-1}, x_i)$ for all $i \in B_t$ and of the regularizer $\nabla_{\mathbf{m}_{t-1}} r(\mathbf{m}_{t-1})$.

        Clip loss gradients: $\mathsf{clip}\left(\nabla_{\mathbf{m}_{t-1}}\ell(\mathbf{m}_{t-1}, x_i)\right) := \frac{\nabla_{\mathbf{m}_{t-1}}\ell(\mathbf{m}_{t-1}, x_i)}{\max\{1, \|\nabla_{\mathbf{m}_{t-1}}\ell(\mathbf{m}_{t-1}, x_i)\|_2\}}$.

        Sample noise $\xi_t \leftarrow \mathcal{N}(0, \sigma^2 I_d)$.

        Update

$$\mathbf{m}_t = \mathbf{m}_{t-1} - \eta \cdot \begin{pmatrix} \sum_{i \in B_t} \mathsf{clip}\left(\nabla_{\mathbf{m}_{t-1}}\ell(\mathbf{m}_{t-1}, x_i)\right) \\ + \nabla_{\mathbf{m}_{t-1}} r(\mathbf{m}_{t-1}) + \xi_t \end{pmatrix}.$$

    **end for**

    **if** `last_iterate_only` **then**

        **return** $\mathbf{m}_T$

    **else if** `intermediate_iterates` **then**

        **return** $\mathbf{m}_0, \mathbf{m}_1, \cdots, \mathbf{m}_{T-1}, \mathbf{m}_T$

    **end if**

**end function**

---

Equation 1 gives us a value of the privacy failure probability parameter $\delta$. But it is more natural to work with the privacy loss bound parameter $\varepsilon$, which can be computed by inverting the formula:

$$\varepsilon_{T,q,\sigma}(\delta) := \min\{\varepsilon \geq 0 : \delta_{T,q,\sigma}(\varepsilon) \leq \delta\}. \tag{3}$$

Both $\delta_{T,q,\sigma}(\varepsilon)$ and $\varepsilon_{T,q,\sigma}(\delta)$ can be computed using existing open-source DP accounting libraries [Goo20]. We also provide a self-contained & efficient method for computing them in Appendix A.

The proof of Theorem 1 is deferred to Appendix A, but we sketch the main ideas: Under the linearity assumption, the output of DP-SGD is just a sum of the gradients and noises. We can reduce to dimension $d = 1$, since the only relevant direction is that of the gradient of the canary[1] (which is constant). We can also ignore the gradients of the other examples. Thus, by rescaling, the worst case pair of output distributions can be represented as in Equation 1. Namely, $Q = \sum_{t=1}^{T} \xi_t$ is simply the noise $\xi_t \leftarrow \mathcal{N}(0, \sigma^2)$ summed over $T$ iterations; this corresponds to the case where the canary is excluded. When the canary is included, it is sampled with probability $q$ in each iteration and thus the total number of times it is sampled over $T$ iterations is $\mathsf{Binomial}(T, q)$. Thus $P$ is the sum of the contributions of the canary and the noise. Finally the definition of differential privacy lets us compute $\varepsilon$ and $\delta$ from this pair of distributions. Tightness follows from the fact that there exists a loss function and pair of inputs such that the corresponding outputs of DP-SGD matches the pair $P$ and $Q$.

---

[1]*The canary* refers to the individual datapoint that is added or removed between neighbouring datasets. This terminology is used in the privacy auditing/attacks literature inspired on the expression "canary in a coalmine."

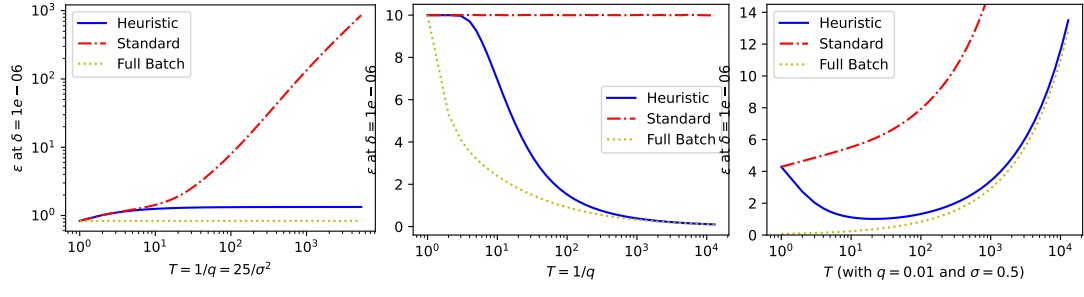

(a) Keep $Tp$ and $T\sigma^2$ constant.     (b) Keep Standard DP constant.     (c) Keep $p$ and $\sigma$ constant.

Figure 1: Comparison of our heuristic to baselines in various parameter regimes. Horizontal axis is number of iterations $T$ and vertical axis is $\varepsilon$ such that we have $(\varepsilon, 10^{-6})$-DP.

### 2.1 Baselines

In addition to privacy auditing, we compare our heuristic to two different baselines in Figure 1. The first is the standard, composition-based analysis. We use the open-source library from Google [Goo20], which computes a tight DP guarantee for DP-SGD with `intermediate_iterates`. Because DP-SGD with `intermediate_iterates` gives the adversary more information than with `last_iterate_only`, this will always give at least as large an estimate for $\varepsilon$ as our heuristic.

We also consider approximating DP-SGD by full batch DP-GD. That is, set $q = 1$ and rescale the learning rate $\eta$ and noise multiplier $\sigma$ to keep the expected step and privacy noise variance constant:

$$\underbrace{\text{DP-SGD}(\mathbf{x}, T, q, \eta, \sigma, \ell, r)}_{\text{batch size} \approx nq,\ T \text{ iterations},\ Tq \text{ epochs}} \approx \underbrace{\text{DP-SGD}(\mathbf{x}, T, 1, \eta \cdot q, \sigma/q, \ell, r)}_{\text{batch size } n,\ T \text{ iterations},\ T \text{ epochs}} . \tag{4}$$

The latter algorithm is full batch DP-GD since at each step it includes each data point in the batch with probability 1. Since full batch DP-GD does not rely on privacy amplification by subsampling, it is much easier to analyze its privacy guarantees. Interestingly, there is no difference between full batch DP-GD with `last_iterate_only` and with `intermediate_iterates`; see Appendix B. Full batch DP-GD generally has better privacy guarantees than the corresponding minibatch DP-SGD and so this baseline usually (but not always) gives smaller values for the privacy leakage $\varepsilon$ than our heuristic. In practice, full batch DP-GD is too computationally expensive to run. But we can use it as an idealized comparison point for the privacy analysis.

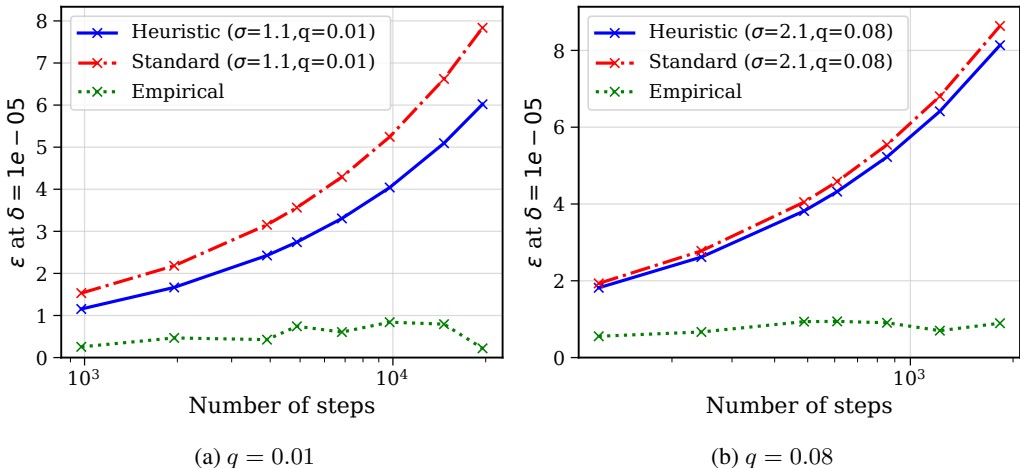

(a) $q = 0.01$          (b) $q = 0.08$

Figure 2: Black-box gradient space attacks fail to achieve tight auditing when other data points are sampled from the data distribution. Heuristic and standard bounds diverge from empirical results, indicating the attack's ineffectiveness. This contrasts with previous work which tightly auditing with access to intermediate updates.

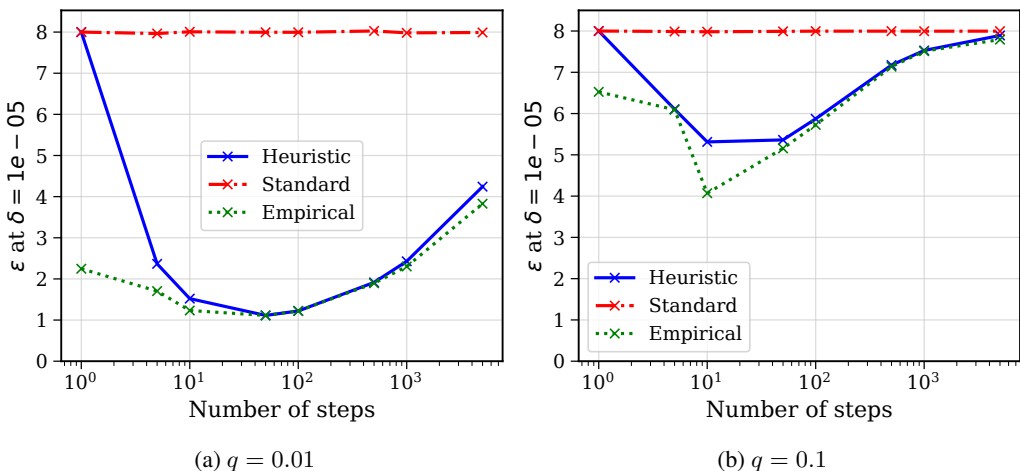

(a) $q = 0.01$

(b) $q = 0.1$

Figure 3: For gradient space attacks with adversarial datasets, the empirical epsilon ($\varepsilon$) closely tracks the final epsilon except for at small step counts, where distinguishing is more challenging. This is evident at both subsampling probability values we study ($q = 0.01$ and $q = 0.1$).

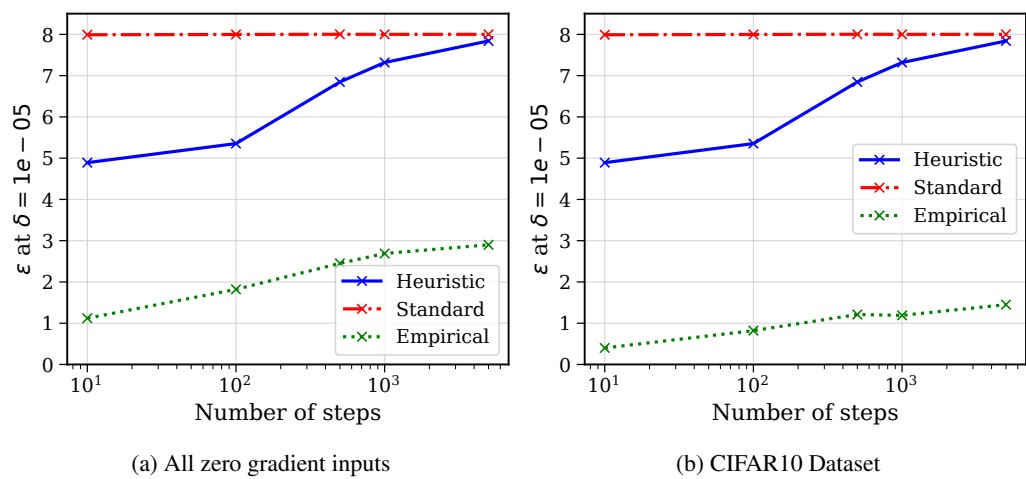

(a) All zero gradient inputs

(b) CIFAR10 Dataset

Figure 4: Input space attacks show promising results with both natural and blank image settings, although blank images have higher attack success. These input space attacks achieve tighter results than gradient space attacks in the natural data setting, in contrast to findings from prior work.

## 3 Empirical Evaluation via Privacy Auditing

**Setup:** We follow the construction of Nasr, Song, Thakurta, Papernot, and Carlini [NSTPC21] where we have 3 entities, adversarial crafter, model trainer, and distinguisher. In this paper, we assume the distinguisher only has access the final iteration of the model parameters. We use the CIFAR10 dataset [Ale09] with a WideResNet model [ZK16] unless otherwise specified; in particular, we follow the training setup of De, Berrada, Hayes, Smith, and Balle [DBHSB22], where we train and audit a model with 79% test accuracy and, using the standard analysis, ($\varepsilon = 8, \delta = 10^{-5}$)-DP. For each experiment we trained 512 CIFAR10 models with and without the canary (1024 total). To compute the empirical lower bounds we use the PLD approach with Clopper-Pearson confidence intervals used by Nasr, Hayes, Steinke, Balle, Tramèr, Jagielski, Carlini, and Terzis [NHSBTJCT23]. Here we assume the adversary knows the sampling rate and the number of iterations and is only estimating the noise multiplier used in DP-SGD, from which the reported privacy parameters ($\varepsilon$ and $\delta$) are derived.

## 3.1 Experimental Results

We implement state-of-the-art attacks from prior work [NSTPC21; NHSBTJCT23]. These attacks heavily rely on the intermediate steps and, as a result, do not achieve tight results. In the next section, we design specific attacks for our heuristic privacy analysis approach to further understand its limitations and potential vulnerabilities. We used Google Cloud `A2-megagpu-16g` machines with 16 Nvidia A100 40GB GPUs. Overall, we use roughly 33,000 GPU hours for our experiments.

**Gradient Space Attack:** The most powerful attacks in prior work are gradient space attacks where the adversary injects a malicious gradient directly into the training process, rather than an example; prior work has shown that this attack can produce tight lower bounds, independent of the dataset and model used for training [NHSBTJCT23]. However, these previous attacks require access to all intermediate training steps to achieve tight results. Here, we use canary gradients in two settings: one where the other data points are non-adversarial and sampled from the real training data, and another where the other data points are designed to have very small gradients ($\approx 0$). This last setting was shown by [NSTPC21] to result in tighter auditing. In all attacks, we assume the distinguisher has access to all adversarial gradient vectors. For malicious gradients, we use Dirac gradient canaries, where gradient vectors consist of zeros in all but a single index. In both cases, the distinguishing test measures the dot product of the final model checkpoint and the gradient canary.

Figure 2 summarizes the results for the non-adversarial data setting, with other examples sampled from the true training data. In this experiment, we fix noise magnitude and subsampling probability, and run for various numbers of training steps. While prior work has shown tight auditing in this setting, we find an adversary without access to intermediate updates obtains much weaker attacks. Indeed, auditing with this strong attack results even in much lower values than the heuristic outputs.

Our other setting assumes the other data points are maliciously chosen. We construct an adversarial "dataset" of $m + 1$ gradients, $m$ of which are zero, and one gradient is constant (with norm equal to the clipping norm), applying gradients directly rather than using any examples. As this experiment does not require computing gradients, it is very cheap to run more trials, so we run this procedure $N = 100,000$ times with the gradient canary, and $N$ times without it, and compute an empirical estimate for $\varepsilon$ with these values. We plot the results of this experiment in Figure 3 together with the $\varepsilon$ output by the theoretical analysis and the heuristic, fixing the subsampling probability and varying the number of update steps. We adjust the noise parameter to ensure the standard theoretical analysis produces a fixed $\varepsilon$ bound. The empirical measured $\varepsilon$ is close to the heuristic $\varepsilon$ except for when training with very small step counts: we expect this looseness to be the result of statistical effects, as lower step counts have higher relative variance at a fixed number of trials.

**Input Space Attack:** In practice, adversaries typically cannot insert malicious gradients freely in training steps. Therefore, we also study cases where the adversary is limited to inserting malicious inputs into the training set. Label flip attacks are one of the most successful approaches used to audit DP machine learning models in prior work [NHSBTJCT23; SNJ23]. For input space attacks, we use the loss of the malicious input as a distinguisher. Similar to our gradient space attacks, we consider two settings for input space attacks: one where other data points are correctly sampled from the dataset, and another where the other data points are blank images.

Figure 4 summarizes the results for this setting. Comparing to Figure 2, input space attacks achieve tighter results than gradient space attacks. This finding is in stark contrast to prior work. The reason is that input space attacks do not rely on intermediate iterates, so they transfer well to our setting.

In all the cases discussed so far, the empirical results for both gradient and input attacks fall below the heuristic analysis and do not violate the upper bounds based on the underlying assumptions. This suggests that the heuristic might serve as a good indicator for assessing potential vulnerabilities. However, in the next section, we delve into specific attack scenarios that exploit the assumptions used in the heuristic analysis to create edge cases where the heuristic bounds are indeed violated.

## 4 Counterexamples

We now test the limits of our heuristic by constructing some artificial counterexamples. That is, we construct inputs to DP-SGD with `last_iterate_only` such that the true privacy loss exceeds the bound given by our heuristic. While we do not expect the contrived structures of these examples to

manifest in realistic learning settings, they highlight the difficulties of formalizing settings where the heuristic gives a provable upper bound on the privacy loss.

## 4.1 Warmup: Zeroing Out The Model Weights

We begin by noting the counterintuitive fact that our heuristic $\varepsilon_{T,q,\sigma}(\delta)$ is *not* always monotone in the number of steps $T$ when the other parameters $\sigma, q, \delta$ are kept constant. This is shown in Figure 1c. More steps means there is both more noise and more signal from the gradients; these effects partially cancel out, but the net effect can be non-monotone.

We can use a regularizer $r(\mathbf{m}) = \|\mathbf{m}\|_2^2 / 2\eta$ so that $\eta \cdot \nabla_{\mathbf{m}} r(\mathbf{m}) = \mathbf{m}$. This regularizer zeros out the model from the previous step, i.e., the update of DP-SGD becomes

$$\mathbf{m}_t = \mathbf{m}_{t-1} - \eta \cdot \left( \sum_{i \in B_t} \mathsf{clip}\left( \nabla_{\mathbf{m}_{t-1}} \ell(\mathbf{m}_{t-1}, x_i) \right) + \nabla_{\mathbf{m}_{t-1}} r(\mathbf{m}_{t-1}) + \xi_t \right) \tag{5}$$

$$= \eta \cdot \sum_{i \in B_t} \mathsf{clip}\left( \nabla_{\mathbf{m}_{t-1}} \ell(\mathbf{m}_{t-1}, x_i) \right) + \xi_t. \tag{6}$$

This means that the last iterate $\mathbf{m}_T$ is effectively the result of only a single iteration of DP-SGD. In particular, it will have a privacy guarantee corresponding to one iteration. Combining this regularizer with a linear loss and a setting of the parameters $T, q, \sigma, \delta$ such that the privacy loss is non-monotone – i.e., $\varepsilon_{T,q,\sigma}(\delta) < \varepsilon_{1,q,\sigma}(\delta)$ – yields a counterexample.

In light of this counterexample, in the next subsection, we benchmark our counterexample against sweeping over smaller values of $T$. I.e., we consider $\max_{t \leq T} \varepsilon_{t,q,\sigma}(\delta)$ instead of simply $\varepsilon_{T,q,\sigma}(\delta)$.

## 4.2 Linear Loss + Quadratic Regularizer

Consider running DP-SGD in one dimension (i.e., $d = 1$) with a linear loss $\ell(\mathbf{m}, x) = \mathbf{m}x$ for the canary and a quadratic regularizer $r(\mathbf{m}) = \frac{1}{2}\alpha\mathbf{m}^2$, where $\alpha \in [0, 1]$ and $x \in [-1, 1]$ and we use learning rate $\eta = 1$. With sampling probability $q$, after $T$ iterations the privacy guarantee is equivalent to distinguishing $Q := \mathcal{N}(0, \widehat{\sigma}^2)$ and $P := \mathcal{N}(\sum_{i \in [T]}(1 - \alpha)^{i-1}\mathsf{Bernoulli}(q), \widehat{\sigma}^2)$, where $\widehat{\sigma}^2 := \sigma^2 \sum_{i \in [T]}(1 - \alpha)^{2(i-1)}$. When $\alpha = 0$, this retrieves linear losses. When $\alpha = 1$, this corresponds to distinguishing $\mathcal{N}(0, \widehat{\sigma}^2)$ and $\mathcal{N}(\mathsf{Bernoulli}(q), \widehat{\sigma}^2)$ or, equivalently, to distinguishing linear losses after $T = 1$ iteration. If we maximize our heuristic over the number of iterations $\leq T$, then our heuristic is tight for the extremes $\alpha \in \{0, 1\}$.

A natural question is whether the worst-case privacy guarantee on this quadratic is always given by $\alpha \in \{0, 1\}$. Perhaps surprisingly, the answer is no: we found that for $T = 3, q = 0.1, \sigma = 1, \alpha = 0$, DP-SGD is $(2.222, 10^{-6})$-DP. For $\alpha = 1$ instead DP-SGD is $(2.182, 10^{-6})$-DP. However, for $\alpha = 0.5$ instead the quadratic loss does not satisfy $(\varepsilon, 10^{-6})$-DP for $\varepsilon < 2.274$.

However, this violation is small, which suggests our heuristic is still a reasonable for this class of examples. To validate this, we consider a set of values for the tuple $(T, q, \sigma)$. For each setting of $T, q, \sigma$, we compute $\max_{t \leq T} \varepsilon_{t,q,\sigma}(\delta)$ at $\delta = 10^{-6}$. We then compute $\varepsilon$ for the linear loss with quadratic regularizer example with $\alpha = 1/2$ in the same setting. Since the support of the random variable $\sum_{i \in [T]}(1 - \alpha)^{i-1}\mathsf{Bernoulli}(q)$ has size $2^T$ for $\alpha = 1/2$, computing exact $\varepsilon$ for even moderate $T$ is computationally intensive. Instead, let $X$ be the random variable equal to $\sum_{i \in [T]}(1 - \alpha)^{i-1}\mathsf{Bernoulli}(q)$, except we round up values in the support which are less than .0005 up to .0005, and then round each value in the support up to the nearest integer power of 1.05. We then compute an exact $\varepsilon$ for distinguishing $\mathcal{N}(0, \widehat{\sigma}^2)$ vs $\mathcal{N}(X, \widehat{\sigma}^2)$. By Lemma 4.5 of Choquette-Choo, Ganesh, Steinke, and Thakurta [CCGST24], we know that distinguishing $\mathcal{N}(0, \widehat{\sigma}^2)$ vs. $\mathcal{N}(\sum_{i \in [T]}(1 - \alpha)^{i-1}\mathsf{Bernoulli}(q), \widehat{\sigma}^2)$ is no harder than distinguishing $\mathcal{N}(0, \widehat{\sigma}^2)$ vs $\mathcal{N}(X, \widehat{\sigma}^2)$, and since we increase the values in the support by no more than 1.05 multiplicatively, we expect that our rounding does not increase $\varepsilon$ by more than 1.05 multiplicatively.

In Figure 5, we plot the ratio of $\varepsilon$ at $\delta = 10^{-6}$ for distinguishing between $\mathcal{N}(0, \widehat{\sigma}^2)$ and $\mathcal{N}(X, \widehat{\sigma}^2)$ divided by the maximum over $i \in [T]$ of $\varepsilon$ at $\delta = 10^{-6}$ for distinguishing between $\mathcal{N}(0, i\sigma^2)$ and $\mathcal{N}(\mathsf{Binomial}(i, q), i\sigma^2)$. We sweep over $T$ and $q$, and for each $q$ In Figure 5a (resp. Figure 5b) we set $\sigma$ such that distinguishing $\mathcal{N}(0, \sigma^2)$ from $\mathcal{N}(\mathsf{Bernoulli}(q), \sigma^2)$ satisfies $(1, 10^{-6})$-DP

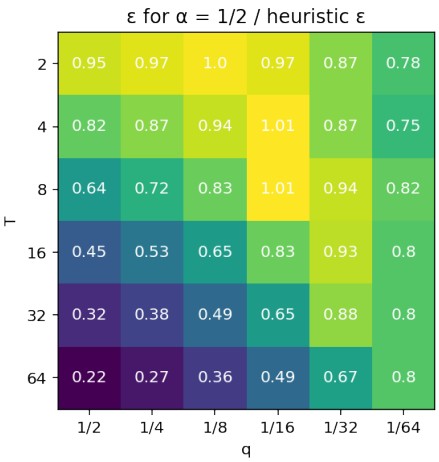
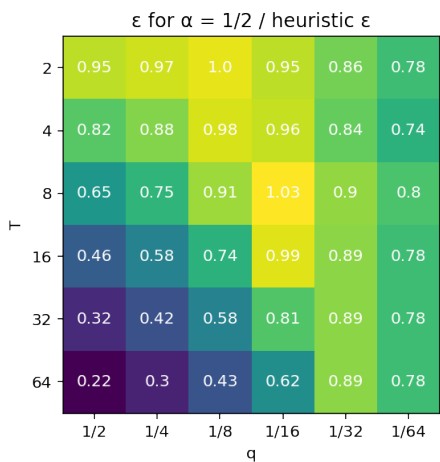

(a) One iteration of DP-SGD satisfies $(1, 10^{-6})$-DP.

(b) One iteration of DP-SGD satisfies $(2, 10^{-6})$-DP.

Figure 5: Ratio of upper bound on $\varepsilon$ for quadratic loss with $\alpha = 0.5$ divided by maximum $\varepsilon$ of $i$ iterations on a linear loss. In Figure 5a (resp. Figure 5b), for each choice of $q$, $\sigma$ is set so 1 iteration of DP-SGD satisfies $(1, 10^{-6})$-DP (resp $(2, 10^{-6})$-DP).

(resp. $(2, 10^{-6})$-DP). In the majority of settings, the linear loss heuristic provides a larger $\varepsilon$ than the quadratic with $\alpha = 1/2$, and even when the quadratic provides a larger $\varepsilon$, the violation is small ($\leq 3\%$). This is evidence that our heuristic is still a good approximation for many convex losses.

## 4.3 Pathological Example

If we allow the regularizer $r$ to be arbitrary – in particular, not even requiring continuity – then the gradient can also be arbitrary. This flexibility allows us to construct a counterexample such that the standard composition-based analysis of DP-SGD with `intermediate_iterates` is close to tight.

Specifically, choose the regularizer so that the update $\mathbf{m}' = \mathbf{m} - \eta \nabla_{\mathbf{m}} r(\mathbf{m})$ does the following: $\mathbf{m}'_1 = 0$ and, for $i \in [d-1]$, $\mathbf{m}'_{i+1} = v \cdot \mathbf{m}_i$. Here $v > 1$ is a large constant. We chose the loss so that, for our canary $x_1$, we have $\nabla_{\mathbf{m}} \ell(\mathbf{m}, x_1) = (1, 0, 0, \cdots, 0)$ and, for all other examples $x_i$ ($i \in \{2, 3, \cdots, n\}$), we have $\nabla_{\mathbf{m}} \ell(\mathbf{m}, x_i) = \mathbf{0}$. Then the last iterate is

$$\mathbf{m}_T = (A_T + \xi_{T,1}, vA_{T-1} + v\xi_{T-1,1} + \xi_{T,2}, v^2 A_{T-2} + v^2 \xi_{T-2,1} + v\xi_{T-1,2} + \xi_{T,3}, \cdots), \quad (7)$$

where $A_t \leftarrow \text{Bernoulli}(p)$ indicates whether or not the canary was sampled in the $t$-th iteration and $\xi_{t,i}$ denotes the $i$-th coordinate of the noise $\xi_t$ added in the $t$-th step. Essentially, the last iterate $\mathbf{m}_T$ contains the history of all the iterates in its coordinates. Namely, the $i$-th coordinate of $\mathbf{m}_T$ gives a scaled noisy approximation to $A_{T-i}$:

$$v^{1-i} \mathbf{m}_{T,i} = A_{T-i} + \sum_{j=0}^{i-1} v^{j+1-i} \xi_{T-j,i-j} \sim \mathcal{N}\left(A_{T-i}, \sigma^2 \frac{1 - v^{-2i}}{1 - v^{-2}}\right). \quad (8)$$

As $v \to \infty$, the variance converges to $\sigma^2$. In other words, if $v$ is large, from the final iterate, we can obtain $\mathcal{N}(A_i, \sigma^2)$ for all $i$. This makes the standard composition-based analysis of DP-SGD tight.

## 4.4 Malicious Dataset Attack

The examples above rely on the regularizer having large unclipped gradients. We now construct a counterexample without a regularizer, instead using other examples to amplify the canary signal.

Our heuristic assumes the adversary does not have access to the intermediate iterations and that the model is linear. However, we can design a nonlinear model and specific training data to directly challenge this assumption. The attack strategy is to use the model's parameters as a sort of noisy storage, saving all iterations within them. Then with access only to the final model, an adversary

Table 1: Previous works showed that large batch sizes achieve high performing models [DBHSB22]. Using our heuristic analysis it is possible to achieve similar performance for smaller batch sizes.

| Batch size | Heuristic $\varepsilon$ | Standard $\varepsilon$ | Accuracy | Empirical $\varepsilon$ |
|---|---|---|---|---|
| 4096 | 6.34 | 8 | 79.5% | 1.7 |
| 512 | 7.0 | 12 | 79.1% | 1.8 |
| 256 | 6.7 | 14 | 79.4% | 1.6 |

can still examine the parameters, extract the intermediate steps, and break the assumption. Our construction introduces a data point that changes its gradient based on the number of past iterations, making it easy to identify if the point was present a given iteration of training. The rest of the data points are maliciously selected to ensure the noise added during training doesn't impact the information stored in the model's parameters. We defer the full details of the attack to Appendix C.

Figure 6 summarizes the results. As illustrated in the figure, this attack achieves a auditing lower bound matching the standard DP-SGD analysis even in the `last_iterate_only` setting. As a result, the attack exceeds our heuristic. However, this is a highly artificial example and it is unlikely to reflect real-world scenarios.

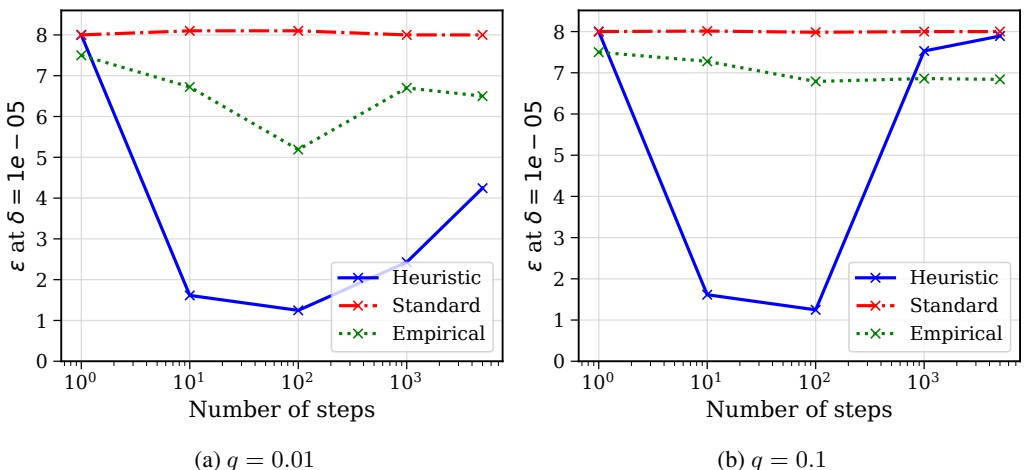

(a) $q = 0.01$            (b) $q = 0.1$

Figure 6: In this adversarial example, the attack encodes all training steps within the final model parameters, thereby violating the specific assumptions used to justify our heuristic analysis.

## 5   Discussion & Conclusion

Both theoretical analysis and privacy auditing are valuable for understanding privacy leakage in machine learning, but each has limitations. Theoretical analysis is inherently conservative, while auditing procedures evaluate only specific attacks, and may thus underrepresent the privacy leakage.

Our work introduces a novel heuristic analysis for DP-SGD that focuses on the privacy implications of releasing only the final model iterate. This approach is based in the empirical observation that linear loss functions accurately model the effectiveness of state of the art membership inference attacks. Our heuristic offers a practical and computationally efficient way to estimate privacy leakage to complement privacy auditing and the standard composition-based analysis of DP-SGD. As shown in Table 1, we trained a series of CIFAR10 models with varying batch sizes that all achieved the similar level of heuristic epsilon, albeit with different standard epsilon values. Remarkably, these models exhibited similar performance and similar empirical epsilon values.

We also acknowledge the limitations of our heuristic by identifying specific counterexamples where the heuristic underestimates the true privacy leakage.

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

| | |
|---|---|
| [DSSUV15] | C. Dwork, A. Smith, T. Steinke, J. Ullman, and S. Vadhan. "Robust traceability from trace amounts". In: *2015 IEEE 56th Annual Symposium on Foundations of Computer Science*. IEEE. 2015, pp. 650–669 (cit. on p. 2). |
| [DWWZK18] | Z. Ding, Y. Wang, G. Wang, D. Zhang, and D. Kifer. "Detecting violations of differential privacy". In: *Proceedings of the 2018 ACM SIGSAC Conference on Computer and Communications Security*. 2018, pp. 475–489 (cit. on p. 2). |
| [FMTT18] | V. Feldman, I. Mironov, K. Talwar, and A. Thakurta. "Privacy amplification by iteration". In: *2018 IEEE 59th Annual Symposium on Foundations of Computer Science (FOCS)*. IEEE. 2018, pp. 521–532. URL: https://arxiv.org/abs/1808.06651 (cit. on p. 2). |
| [Goo20] | Google. *Differential Privacy Accounting*. https://github.com/google/differential-privacy/tree/main/python/dp_accounting. 2020 (cit. on pp. 3, 4). |
| [HSRDTMPSNC08] | N. Homer, S. Szelinger, M. Redman, D. Duggan, W. Tembe, J. Muehling, J. V. Pearson, D. A. Stephan, S. F. Nelson, and D. W. Craig. "Resolving individuals contributing trace amounts of DNA to highly complex mixtures using high-density SNP genotyping microarrays". In: *PLoS genetics* 4.8 (2008), e1000167 (cit. on p. 2). |
| [JUO20] | M. Jagielski, J. Ullman, and A. Oprea. "Auditing differentially private machine learning: How private is private sgd?" In: *Advances in Neural Information Processing Systems* 33 (2020), pp. 22205–22216 (cit. on p. 2). |
| [KJH20] | A. Koskela, J. Jälkö, and A. Honkela. "Computing tight differential privacy guarantees using fft". In: *International Conference on Artificial Intelligence and Statistics*. PMLR. 2020, pp. 2560–2569. URL: https://arxiv.org/abs/1906.03049 (cit. on p. 1). |
| [LF20] | K. Leino and M. Fredrikson. "Stolen memories: Leveraging model memorization for calibrated {White-Box} membership inference". In: *29th USENIX security symposium (USENIX Security 20)*. 2020, pp. 1605–1622. URL: https://arxiv.org/abs/1906.11798 (cit. on p. 2). |
| [Mir17] | I. Mironov. "Rényi differential privacy". In: *2017 IEEE 30th computer security foundations symposium (CSF)*. IEEE. 2017, pp. 263–275. URL: https://arxiv.org/abs/1702.07476 (cit. on p. 1). |
| [NHSBTJCT23] | M. Nasr, J. Hayes, T. Steinke, B. Balle, F. Tramèr, M. Jagielski, N. Carlini, and A. Terzis. "Tight Auditing of Differentially Private Machine Learning". In: *arXiv preprint arXiv:2302.07956* (2023). URL: https://arxiv.org/abs/2302.07956 (cit. on pp. 1, 2, 5, 6). |
| [NSTPC21] | M. Nasr, S. Song, A. Thakurta, N. Papernot, and N. Carlini. "Adversary instantiation: Lower bounds for differentially private machine learning". In: *2021 IEEE Symposium on security and privacy (SP)*. IEEE. 2021, pp. 866–882. URL: https://arxiv.org/abs/2101.04535 (cit. on pp. 1, 2, 5, 6). |
| [SDSOJ19] | A. Sablayrolles, M. Douze, C. Schmid, Y. Ollivier, and H. Jégou. "White-box vs black-box: Bayes optimal strategies for membership inference". In: *International Conference on Machine Learning*. PMLR. 2019, pp. 5558–5567. URL: https://arxiv.org/abs/1908.11229 (cit. on p. 2). |
| [SNJ23] | T. Steinke, M. Nasr, and M. Jagielski. "Privacy auditing with one (1) training run". In: *Advances in Neural Information Processing Systems* 36 (2023). URL: https://arxiv.org/abs/2305.08846 (cit. on pp. 2, 6). |
| [SOJH09] | S. Sankararaman, G. Obozinski, M. I. Jordan, and E. Halperin. "Genomic privacy and limits of individual detection in a pool". In: *Nature genetics* 41.9 (2009), pp. 965–967 (cit. on p. 2). |
| [SSSS17] | R. Shokri, M. Stronati, C. Song, and V. Shmatikov. "Membership inference attacks against machine learning models". In: *2017 IEEE symposium on security and privacy (SP)*. IEEE. 2017, pp. 3–18 (cit. on p. 2). |
| [Ste22] | T. Steinke. "Composition of Differential Privacy & Privacy Amplification by Subsampling". In: *arXiv preprint arXiv:2210.00597* (2022). URL: https://arxiv.org/abs/2210.00597 (cit. on p. 1). |

| | |
|---|---|
| [TTSSJC22] | F. Tramer, A. Terzis, T. Steinke, S. Song, M. Jagielski, and N. Carlini. "Debugging differential privacy: A case study for privacy auditing". In: *arXiv preprint arXiv:2202.12219* (2022). URL: https://arxiv.org/abs/2202.12219 (cit. on p. 2). |
| [WBKBGGG23] | Y. Wen, A. Bansal, H. Kazemi, E. Borgnia, M. Goldblum, J. Geiping, and T. Goldstein. "Canary in a coalmine: Better membership inference with ensembled adversarial queries". In: *ICLR*. 2023. URL: https://arxiv.org/abs/2210.10750 (cit. on p. 2). |
| [YS22] | J. Ye and R. Shokri. "Differentially private learning needs hidden state (or much faster convergence)". In: *Advances in Neural Information Processing Systems* 35 (2022), pp. 703–715. URL: https://arxiv.org/abs/2203.05363 (cit. on p. 2). |
| [ZBWTSRPNK22] | S. Zanella-Béguelin, L. Wutschitz, S. Tople, A. Salem, V. Rühle, A. Paverd, M. Naseri, and B. Köpf. "Bayesian estimation of differential privacy". In: *arXiv preprint arXiv:2206.05199* (2022) (cit. on p. 2). |
| [ZK16] | S. Zagoruyko and N. Komodakis. "Wide residual networks". In: *arXiv preprint arXiv:1605.07146* (2016) (cit. on pp. 5, 20). |
| [ZLS23] | S. Zarifzadeh, P. C.-J. M. Liu, and R. Shokri. "Low-Cost High-Power Membership Inference by Boosting Relativity". In: (2023). URL: https://arxiv.org/abs/2312.03262 (cit. on p. 2). |

# A    Proof of Theorem 1

*Proof.* Let $x_{i^*}$ be the canary, let $D$ be the dataset with the canary and $D'$ be the dataset without the canary. Since $\ell$ and $r$ are linear, wlog we can assume $r = 0$ and $\nabla_{\mathbf{m}_{t-1}}\ell(\mathbf{m}_{t-1}, x_i) = \mathbf{v}_i$ for some set of vectors $\{\mathbf{v}_i\}$, such that $\|\mathbf{v}_i\|_2 \leq 1$. We can also assume wlog $\|\mathbf{v}_{i^*}\| = 1$ since, if $\|\mathbf{v}_{i^*}\| < 1$, the final privacy guarantee we show only improves.

We have the following recursion for $\mathbf{m}_t$:

$$\mathbf{m}_t = \mathbf{m}_{t-1} - \eta\left(\sum_{i \in B_t} \mathbf{v}_i + \xi_t\right), \qquad \xi_t \stackrel{i.i.d}{\sim} \mathcal{N}(0, \sigma^2 I_d).$$

Unrolling the recursion:

$$\mathbf{m}_t = \mathbf{m}_0 - \eta\left[\sum_{t \in [T]} \sum_{i \in B_t} \mathbf{v}_i + \xi\right], \qquad \xi \sim N(0, T\sigma^2 I_d).$$

By the post-processing property of DP, we can assume that in addition to the final model $\mathbf{m}_T$, we release $\mathbf{m}_0$ and $\{B_t \setminus \{x_{i^*}\}\}_{t \in [T]}$, that is we release all examples that were sampled in each batch except for the canary. The following $f$ is a bijection, computable by an adversary using the released information:

$$f(\mathbf{m}_T) := -\left[\frac{\mathbf{m}_T - \mathbf{m}_0}{\eta} - \sum_{t \in [T]} \sum_{i \in B_t \setminus \{x_{i^*}\}} \mathbf{v}_i.\right]$$

Since $f$ is a bijection, distinguishing $\mathbf{m}_T$ sampled using $D$ and $D'$ is equivalent to distinguishing $f(\mathbf{m}_T)$ instead. Now we have $f(\mathbf{m}_T) = \mathcal{N}(0, T\sigma^2 I_d)$ for $D'$, and $f(\mathbf{m}_T) = \mathcal{N}(0, T\sigma^2 I_d) + k\mathbf{v}_{i^*}$, $k \sim \text{Binomial}(T, q)$. For any vector $\mathbf{u}$ orthogonal to $\mathbf{v}_{i^*}$, by isotropy of the Gaussian distribution the distribution of $\langle f(\mathbf{m}_T), \mathbf{u}\rangle$ is the same for both $D$ and $D'$ and independent of $\langle f(\mathbf{m}_T), \mathbf{v}_{i^*}\rangle$, hence distinguishing $f(\mathbf{m}_T)$ given $D$ and $D'$ is the same as distingushing $\langle f(\mathbf{m}_T), \mathbf{v}_{i^*}\rangle$ given $D$ and $D'$. Finally, the distribution of $\langle f(\mathbf{m}_T), \mathbf{v}_{i^*}\rangle$ is exactly $P$ for $D$ and exactly $Q$ for $D'$. By post-processing, this gives the theorem.

We can also see that the function $\delta_{T,q,\sigma}$ is tight (i.e., even if we do not release $B_t \setminus \{x_{i^*}\}$), by considering the 1-dimensional setting, where $\mathbf{v}_i = 0$ for $i \neq i^*$ and $\mathbf{v}_{i^*} = -1, \eta = 1, \mathbf{m}_0 = 0$. Then, the distribution of $\mathbf{m}_T$ given $D$ is exactly $P$, and given $D'$ is exactly $Q$. $\qquad\square$

## A.1 Computing $\delta$ from $\varepsilon$

Here, we give an efficiently computable expression for the function $\delta_{T,q,\sigma}(\varepsilon)$. Using $P, Q$ as in Theorem 1, let $f(y)$ be the privacy loss for the output $y$:

$$f(y) = \log\left(\frac{P(y)}{Q(y)}\right) = \log\left(\sum_{k=0}^{T}\binom{T}{k}q^k(1-q)^{n-k}\frac{\exp(-(y-k)^2/2T\sigma^2)}{\exp(-y^2/2T\sigma^2)}\right)$$

$$= \log\left(\sum_{k=0}^{T}\binom{T}{k}q^k(1-q)^k\exp\left(\frac{2ky-k^2}{2T\sigma^2}\right)\right).$$

Then for any $\varepsilon$, using the fact that $S = \{y : f(y) \geq \varepsilon\}$ maximizes $P(S) - e^\varepsilon Q(S)$, we have:

$$
\begin{aligned}
H_{e^\varepsilon}(P, Q) &= P(\{y : f(y) \geq \varepsilon\}) - e^\varepsilon Q(\{y : f(y) \geq \varepsilon\}) \\
&= P(\{y : y \geq f^{-1}(\varepsilon)\}) - e^\varepsilon Q(\{y : y \geq f^{-1}(\varepsilon)\}) \\
&= \sum_{k=0}^{T}\binom{T}{k}q^k(1-q)^k\Pr[\mathcal{N}(k, T\sigma^2) \geq f^{-1}(\varepsilon)] - e^\varepsilon \Pr[\mathcal{N}(0, T\sigma^2) \geq f^{-1}(\varepsilon)].
\end{aligned}
$$

Similarly, $S = \{y : f(y) \leq -\varepsilon\}$ maximizes $Q(S) - e^\varepsilon P(S)$ so we have:

$$
\begin{aligned}
H_{e^\varepsilon}(Q, P) &= Q(\{y : f(y) \leq -\varepsilon\}) - e^\varepsilon P(\{y : f(y) \leq -\varepsilon\}) \\
&= Q(\{y : y \leq f^{-1}(-\varepsilon)\}) - e^\varepsilon P(\{y : y \leq f^{-1}(-\varepsilon)\}) \\
&= \Pr[\mathcal{N}(0, T\sigma^2) \leq f^{-1}(-\varepsilon)] - e^\varepsilon \sum_{k=0}^{T}\binom{T}{k}q^k(1-q)^k\Pr[\mathcal{N}(k, T\sigma^2) \leq f^{-1}(-\varepsilon)].
\end{aligned}
$$

These expressions can be evaluated efficiently. Since $f$ is monotone, it can be inverted via binary search. We can also use binary search to evaluate $\varepsilon$ as a function of $\delta$.

## B Linear Worst Case for Full Batch Setting

It turns out that in the full-batch setting, the worst-case analyses of DP-GD with `intermediate_iterates` and with `last_iterate_only` are the same. This phenomenon arises because there is no subsampling (because $q = 1$ in Algorithm 1) and thus the algorithm is "just" the Gaussian mechanism. Intuitively, DP-GD with `intermediate_iterates` corresponds to $T$ calls to the Gaussian mechanism with noise multiplier $\sigma$, while DP-GD with `last_iterate_only` corresponds to one call to the Gaussian mechanism with noise multiplier $\sigma/\sqrt{T}$; these are equivalent by the properties of the Gaussian distribution.

We can formalize this using the language of Gaussian DP [DRS19]: DP-GD (Algorithm 1 with $q = 1$) satisfies $\sqrt{T}/\sigma$-GDP. (Each iteration satisfies $1/\sigma$-GDP and adaptive composition implies the overall guarantee.) This means that the privacy loss is exactly dominated by that of the Gaussian mechanism with noise multiplier $\sigma/\sqrt{T}$. Linear losses give an example such that DP-GD with `last_iterate_only` has exactly this privacy loss, since the final iterate reveals the sum of all the noisy gradient estimates. The worst-case privacy of DP-GD with `intermediate_iterates` is no worse than that of DP-GD with `last_iterate_only`. The reverse is also true (by postprocessing).

In more detail: For $T$ iterations of (full-batch) DP-GD on a linear losses, if the losses are (wlog) 1-Lipschitz and we add noise $\mathcal{N}(0, \frac{\sigma^2}{n^2} \cdot I)$ to the gradient in every round, distinguishing the last

iterate of DP-SGD on adjacent databases is equivalent to distinguishing $\mathcal{N}(0, T\sigma^2)$ and $\mathcal{N}(T, T\sigma^2)$. This can be seen as a special case of Theorem 1 for $p = 1$, so we do not a give a detailed argument here.

If instead we are given every iteration $\mathbf{m}_t$, for *any* 1-Lipschitz loss, distinguishing the joint distributions of $\mathbf{m}_t$ given $\mathbf{m}_{t-1}$ on adjacent databases is equivalent to distinguishing $\mathcal{N}(0, \sigma^2)$ and $\mathcal{N}(1, \sigma^2)$. In turn, distinguishing the distribution of all iterates on adjacent databases is equivalent to distinguishing $\mathcal{N}(\mathbf{0}^T, \sigma^2 I_T)$ and $\mathcal{N}(\mathbf{1}^T, \sigma^2 I_T)$, where $\mathbf{0}^T$ and $\mathbf{1}^T$ are the all-zeros and all-ones vectors in $\mathbb{R}^T$. Because the Gaussian distribution is isotropic, distinguishing $\mathcal{N}(\mathbf{0}^T, \sigma^2 I_T)$ and $\mathcal{N}(\mathbf{1}^T, \sigma^2 I_T)$ is equivalent to distinguishing $\langle \mathbf{x}, \mathbf{1}^T \rangle$ where $\mathbf{x} \sim \mathcal{N}(\mathbf{0}^T, \sigma^2 I_T)$ and $\langle \mathbf{x}, \mathbf{1}^T \rangle$ where $\mathbf{x} \sim \mathcal{N}(\mathbf{1}^T, \sigma^2 I_T)$. These distributions are $\mathcal{N}(0, T\sigma^2)$ and $\mathcal{N}(T, T\sigma^2)$, the exact pair of distributions we reduced to for last-iterate analysis of linear losses.

## C  Malicious Dataset Attack Details

Algorithms 2, 3, and 4 summarizes the construction for the attack. The attack assume the model parameters have dimension equal to the number of iterations. It also assumes each data point can reference which iteration of training is currently happening (this can be implemented by having a single model parameter which increments in each step, independently of the training examples, without impacting the privacy of the training process). Then we build our two datasets $D$ and $D' = D \cup \{x\}$ so that all points in dataset $D$ ("repeaters") run Algorithm 3 to compute gradients and the canary point in $D'$ runs Algorithm 2 to compute its gradient. Our attack relies heavily on DP-SGD's lack of assumptions on the data distribution and any specific properties of the model or gradients. Algorithm 2, which generates the canary data point, is straightforward. Its goal is to store in the model parameters whether it was present in iteration $i$ by outputting a gradient that changes only the $i$-th index of the model parameters by 1 (assuming a clipping threshold of 1).

All other data points, the "repeaters", are present in both datasets ($D$ and $D'$), and have three tasks:

- Cancel out any noise added to the model parameters at an index larger than the current iteration. At iteration $i$, their gradients for parameters from index $i$ onward will be the same as the current value of the model parameter, scaled by the batch size and the learning rate to ensure this parameter value will be 0 after the update.

- Evaluate whether the canary point was present in the previous iteration by comparing the model parameter at index $i - 1$ with a threshold, and rewrite the value of that model parameter to a large value if the canary was present.

- Ensure that all previous decisions are not overwritten by noise by continuing to rewrite them with a large value based on their previous value.

To achieve all of these goals simultaneously, we require that the batch size is large enough that the repeaters' updates are not clipped.

Finally Algorithm 4 runs DP-SGD, with repeater points computing gradients with Algorithm 3 and the canary point, sampled with probability $p$, computing its gradient using Algorithm 2. In our experiments we run Algorithm 4 100,000 times. And to evaluate if the model parameters was from dataset $D$ or $D'$ we run a hypothesis test on the values of the model parameters. All constants are chosen to ensure all objectives of the repeaters are satisfied.

---

**Algorithm 2** Canary data point

---

1: **function** ADV($\mathbf{x}$, $i$)
2:     Initialize $\mathbf{a}$ as a zero vector of the same dimension as $\mathbf{x}$
3:     Set $a_i \leftarrow 1$                         ▷ Set the $i$-th component to 1
4:     **return** $-\mathbf{a}$
5: **end function**

---

**Algorithm 3** Additional data points

---

**Require:** model parameters $\mathbf{x}$, iteration number $i$, batch size $N$, learning rate $\eta$, previous history threshold $t_{\text{past}}$, last iteration threshold $t_{\text{last}}$, history amplification value BIG_VAL

1: **function** REPEATERS($\mathbf{x}$, $i$, $N$, $\eta$, $t_{\text{past}}$, $t_{\text{last}}$, BIG_VAL)
2:     $\mathbf{h} \leftarrow \mathbf{x}_{0:i}$                   $\triangleright$ Parameter "history" up to iteration $i$, not inclusive
3:     $\mathbf{f} \leftarrow \mathbf{x}_{i:end}$                  $\triangleright$ Future and current parameters, starting from iteration $i$
4:     $\mathbf{f} \leftarrow -\mathbf{f}/(\eta \cdot N)$                 $\triangleright$ Remove noise from last iteration
5:     base_history $\leftarrow -\mathbf{x}_{0:i}/(\eta \cdot N)$         $\triangleright$ By default, zero out entire history
6:     **if** length($\mathbf{h}$) $> 1$ **then**
7:         $\mathbf{h}_{0:i-1} \leftarrow \text{BIG\_VAL}/(\eta \cdot N) \cdot (2\mathbb{1}[\mathbf{h}_{0:i-1} \geq t_{\text{past}}] - 1)$     $\triangleright$ If an old iteration is large enough, it was a canary iteration, so amplify it
8:     **end if**
9:     **if** length($\mathbf{h}$) $> 0$ **then**
10:        $\mathbf{h}_{i-1} \leftarrow \text{BIG\_VAL}/(\eta \cdot N) \cdot (2\mathbb{1}[\mathbf{h}_i \geq t_{\text{last}}] - 1)$ $\triangleright$ If the last iteration is large enough, it was a canary iteration, so amplify it
11:     **end if**
12:     $\mathbf{h} \leftarrow \mathbf{h} + \text{base\_history}$               $\triangleright$ Don't zero out canary iterations
13:     $\mathbf{a} \leftarrow \text{concatenate}(\mathbf{h}, \mathbf{f})$
14:     **return** $-\mathbf{a}$
15: **end function**

---

**Algorithm 4** Encoding Attacking

---

**Require:** add-diff, whether to add the canary, batch size $N$, sampling rate $p$, learning rate ($\eta$), iteration count/parameter count $D$

1: **function** RUN_DPSGD(add-diff)
2:     $C \leftarrow 1$
3:     Initialize model $\mathbf{m} \leftarrow \mathbf{0}$ of dimension $D$
4:     **for** $i = 0$ to $D$ **do**
5:         Generate a uniform random value $q \in [0, 1]$
6:         $\mathbf{r} \leftarrow \text{repeaters}(\mathbf{m}, i)$
7:         Compute norm $c \leftarrow ||\mathbf{r}||$
8:         **if** $c > 0$ **then**
9:             Normalize $\mathbf{r} \leftarrow \mathbf{r}/\max(c, C)$
10:        **end if**
11:        Adjusted vector $\mathbf{z} \leftarrow \mathbf{r} \times N$
12:        Verify condition on $\mathbf{m}_i$
13:        **if** $p \leq q$ and add-diff **then**
14:            $\mathbf{r} \leftarrow \text{adv}(\mathbf{m}, i)$
15:            Normalize and update $\mathbf{z}$
16:        **end if**
17:        Apply Gaussian noise to $\mathbf{z}$
18:        Update model $\mathbf{m} \leftarrow \mathbf{m} - \mathbf{z} \times \eta$
19:     **end for**
20:     **return** $\mathbf{m}$
21: **end function**

---

