# OpenReview forum: "The Last Iterate Advantage: Empirical Auditing and Principled Heuristic Analysis of Differentially Private SGD"
_NeurIPS.cc/2024/Conference — Submitted to NeurIPS 2024_

### Official Review · Reviewer_buwF · 2024-07-07

**Soundness:** 2
**Presentation:** 3
**Contribution:** 2
**Rating:** 3
**Confidence:** 3

**Summary:**

The paper presents a heuristic approach for evaluating the privacy of DP-SGD when only the last model iteration is released. This method contrasts with traditional analyses that consider all intermediate updates, offering a more practical assessment for scenarios where adversaries only access the final model. The proposed heuristic is experimentally shown to provide reliable privacy leakage estimates, making it a valuable tool for pre-audit privacy assessments.

**Strengths:**

1. Focus on a good and important question.
2. Good explanation and clear paper layout.
3. Propose a new analysis neither from the theoretical nor empirical point of view.

**Weaknesses:**

I think the proposed method is interesting and new but I still have some questions.

1. I know the linear loss function assumption is common in theoretical analysis but it seems that the proposed method wants to have contributions in the empirical case, so why still make the linear assumption?

2. While I appreciate the effort to introduce a Heuristic analysis, I remain skeptical about its necessity and effectiveness. The primary benefit of theoretical analysis is its precision and rigor, which often include the flexibility to adjust bounds as needed. If the goal is to find a more relaxed lower bound on privacy risks, this can often be achieved by simply loosening the constraints within the existing theoretical framework. Introducing a separate heuristic analysis seems to complicate matters without providing clear advantages.

3. I do not think you are using a correct baseline. When you make that only the last iteration model can be seen assumption, it is not fair to use normal DP-SGD analysis. I think it is better to use the theoretical analysis from those hidden state papers you cited. I am curious if you compare your proposed method with those methods, will you still get the same conclusion?

4. I find Table 1 in the paper somewhat unclear and would appreciate further explanation from the authors regarding its purpose and implications. The table suggests that similar levels of heuristic ε are achieved across varying batch sizes, yet there is a noticeable increase in the standard privacy budget for smaller batches to maintain comparable performance. This observation seems to underscore the well-known impact of batch size rather than demonstrating an advantage of the proposed heuristic method.
Could the authors elaborate on how this data relates to the efficacy of the heuristic analysis?

**Questions:**

Please check the weaknesses.

**Limitations:**

Please check the weaknesses.

---

> ### Author Rebuttal · Authors · 2024-08-06
>
> We thank the reviewer for their time and comments.
>
> > 1. I know the linear loss function assumption is common in theoretical analysis but it seems that the proposed method wants to have contributions in the empirical case, so why still make the linear assumption?
>
> Unfortunately, we do not understand this question. We propose a heuristic. It is theoretically justified in the linear case, but we argue that it is still representative for real deep learning settings.
>
> > 2. While I appreciate the effort to introduce a Heuristic analysis, I remain skeptical about its necessity and effectiveness. The primary benefit of theoretical analysis is its precision and rigor, which often include the flexibility to adjust bounds as needed. If the goal is to find a more relaxed lower bound on privacy risks, this can often be achieved by simply loosening the constraints within the existing theoretical framework. Introducing a separate heuristic analysis seems to complicate matters without providing clear advantages.
>
> It is not clear how we can loosen the constraints within the existing theoretical framework. One natural approach is to make some kind of assumption that makes the privacy analysis easier; this is precisely what we do.
>
> > 3. I do not think you are using a correct baseline. When you make that only the last iteration model can be seen assumption, it is not fair to use normal DP-SGD analysis. I think it is better to use the theoretical analysis from those hidden state papers you cited. I am curious if you compare your proposed method with those methods, will you still get the same conclusion?
>
> It is difficult to compare to these prior works [[FMTT18](https://arxiv.org/abs/1808.06651); [CYS21](https://arxiv.org/abs/2102.05855); [YS22](https://arxiv.org/abs/2203.05363); [AT22](https://arxiv.org/abs/2205.13710); [BSA24](https://arxiv.org/abs/2403.00278)] for two reasons. First, many of these prior works make an assumption of strong convexity, which is technically incompatible with our linearity assumption. If we try to compare anyway, how would we set the strong convexity parameter for a fair comparison? We could always set the strong convexity parameter in such a way that our numbers are better. Second, most of these prior works are theoretical in nature and it is nontrivial to extract concrete numbers from their theorems to compare with.
>
> > 4. I find Table 1 in the paper somewhat unclear and would appreciate further explanation from the authors regarding its purpose and implications. The table suggests that similar levels of heuristic $\varepsilon$ are achieved across varying batch sizes, yet there is a noticeable increase in the standard privacy budget for smaller batches to maintain comparable performance. This observation seems to underscore the well-known impact of batch size rather than demonstrating an advantage of the proposed heuristic method. Could the authors elaborate on how this data relates to the efficacy of the heuristic analysis?
>
> The accepted wisdom, e.g. [[DBHSB22](https://arxiv.org/abs/2204.13650)], is that larger batch size yields better privacy/utility with DP-SGD. However, this is based on the standard DP-SGD analysis. Table 1 questions this accepted wisdom. Our heuristic, like the results of privacy auditing, is relatively insensitive to the batch size.
>
> Training with large batch sizes is computationally costly, so if it is not truly improving privacy, we shouldn’t advise practitioners to do it. Further research is needed to understand whether or not larger batch sizes are truly beneficial.
>
> We hope that we have addressed the reviewers concerns and that they reconsider their recommendation.

---

> > ### Comment · Reviewer_buwF · 2024-08-08
> >
> > Thank you for your response. Let me clarify what I mean for the first question.
> > Is it feasible to apply your proposed method to a non-linear loss like a cross-entropy loss that the community usually uses? I also wonder why you think a technique theoretically justified in the linear case can be representative of real deep learning. For example, I want to train a Vision transformer model on ImageNet with DP-SGD. Do you think it is possible to only use a linear loss? Or in other words, will your proposed method can be applied in this real deep learning case?

---

> ### Author Response · Authors · 2024-08-08
>
> Thank you for the clarification.
>
> >  Is it feasible to apply your proposed method to a non-linear loss like a cross-entropy loss that the community usually uses?
>
> Realistic deep learning losses yield complicated output distributions on the model weights and, unfortunately, it is not possible to give an exact differential privacy analysis.
> This is why there is a lot of work on privacy auditing, which aims to empirically approximate an exact DP analysis. Thus we compare our heuristic to state-of-the-art privacy auditing methods.
>
> In Section 4.2 we performed an exact analysis of quadratic losses. This is slightly more general than linear losses. But even here the output distribution becomes a mixture of Gaussians which is nontrivial to analyze.
>
> > I also wonder why you think a technique theoretically justified in the linear case can be representative of real deep learning.
>
> It is indeed surprising that linear losses yield a good approximation to real deep learning in terms of privacy.
> The key observation that inspired this paper is that, in the practice of privacy auditing, the strongest membership inference results are achieved by examples ("canaries") with gradients that are the same in each iteration of SGD. This corresponds exactly to the case of a linear loss function.
>
> So our linear approximation is well-justified empirically. Theoretically, Appendix B shows that *in the full batch setting* our linear approximation is indeed the worst case, although this proof does not extend to the minibatch setting.
>
> > For example, I want to train a Vision transformer model on ImageNet with DP-SGD. Do you think it is possible to only use a linear loss? Or in other words, will your proposed method can be applied in this real deep learning case?
>
> Obviously, training with a linear loss would result in non-convergence of DP-SGD, so we don't recommend doing that. :-)
>  Our thesis is that, in the last iterate setting, for the privacy analysis, we can *pretend* that we have a linear loss to give a heuristic privacy analysis. This heuristic privacy analysis is optimistic, but it may be closer to the "true" privacy loss than the standard pessimistic DP-SGD analysis. We argue that this is a useful/interesting perspective to add to practical private deep learning, although it still leaves open many questions.

---

> > ### Comment · Reviewer_buwF · 2024-08-13
> >
> > Thanks authors for the detailed explanations. I have read the rebuttal and comments but not be fully convinced by them. I will keep the same score.

---

### Official Review · Reviewer_8UgB · 2024-07-12

**Soundness:** 2
**Presentation:** 3
**Contribution:** 3
**Rating:** 3
**Confidence:** 3

**Summary:**

This paper proposes a heuristic privacy analysis of releasing only the final model iterate of differentailly private gradient descent (DP-SGD). The analysis is based off of the worst-case differential privacy guarantee of DP-SGD with linear losses, under the assumption that the heuristic can be applied to more general loss functions in order to approximate the privacy loss.

**Strengths:**

* The premise of the paper (a heuristic privacy analysis of releasing only the final model iterate of DP-SGD) is very interesting, and Theorem 1 is a cool result.
* The paper thoroughly assesses the limitations of the heuristic (in Section 4).

**Weaknesses:**

* I don’t know how useful the heuristic analysis would be in practice — beyond a lightweight sanity check — since ultimately it is just a heuristic and not a rigorous upper or lower bound on the privacy loss.

* The empirical study of the heuristic looks to be very thorough, but sparse on interpretation. I would have appreciated more discussion on the figures, and didn’t really feel like there was a strong take-home message from the paper.

* Algorithm 1 is DP-SGD with a regularizer, but in practice it is somewhat rare to use explicit regularization with DP-SGD. So I’m not sure that the heuristic would be widely applicable to the more common implementation of DP-SGD without regularization.

**Questions:**

1. One of the potential use cases of the heuristic (lines 82-85) is to predict the outcome of privacy auditing. So I’m wondering:
   * If one of the advantages of the heuristic is that it’s less computationally expensive than privacy auditing, but you’ll have to do the computationally expensive privacy auditing anyway (in order to compare), then what is the point?
   * One of the disadvantages of privacy auditing is that it’s difficult to perform correctly, but the heuristic doesn’t have rigorous theoretical guarantees. If privacy auditing and the heuristic don’t agree, then how would you know which one was wrong?

2. In Figure 3, why does the heuristic $\epsilon$ scale with the number of iterations $T$? Since the heuristic is a last-iterate analysis, I would have imagined that it would converge to a constant?

3. Would it be possible to extend Theorem 1 to non-regularized DP-SGD?

4. It’s not obvious to me why the privacy guarantee for an arbitrary loss would be similar to the privacy guarantee for a linear loss (and not even a GLM loss, mind you). Maybe I missed this in the paper, but would it be possible to further justify why the heuristic would be a good approximation in cases where the linear loss assumption doesn’t hold?

5. I feel like the proof of Theorem 1 is hard to appreciate without knowing more technical preliminaries, such as how the hockey-stick divergence can be used to show that an algorithm satisfies DP. I wonder if the proof could be re-framed in the language of privacy profiles and dominating pairs? (https://arxiv.org/abs/1807.01647 and https://arxiv.org/abs/2106.08567)

**Limitations:**

The authors have adequately addressed the limitations of their work.

---

> ### Author Rebuttal · Authors · 2024-08-06
>
> We thank the reviewer for their thoughtful comments. We address their main concerns:
>
> > * I don’t know how useful the heuristic analysis would be in practice.
>
> We believe that such a “sanity check” is useful in practice. In practice, often the standard $\varepsilon$ DP parameter is uncomfortably large and we seek additional validation through methods like privacy auditing. Our heuristic can provide additional validation, which is particularly useful because it is easy to do privacy auditing incorrectly, which gives a false sense of security.
>
> More importantly, beyond immediate practical impact, we hope that our work leads to further scientific exploration of the privacy implications of the last iterate setting. In other words, our work identifies a phenomenon and proposes an explanation (and also challenges that explanation). Thus we believe that our work is valuable as basic science, even if the heuristic is not immediately useful.
>
> > * The empirical study of the heuristic looks to be very thorough, but sparse on interpretation. I would have appreciated more discussion on the figures, and didn’t really feel like there was a strong take-home message from the paper.
>
> Space permitting, we will add further discussion of the experimental results in the revision (the camera ready allows an extra page of content).
>
> Overall, we attempted to present the results in a neutral manner, to let readers draw their own conclusions. In particular, we put a lot of effort into probing the limitations of our heuristic. (E.g. Section 4 shows that our heuristic can be too optimistic and Figure 2 shows that it doesn’t fully explain the gap between theoretical analyses and empirical results.)
>
> The take home message is (i) that the last iterate setting is interesting and there is a gap between the upper bounds we can prove and the lower bounds we get empirically; and (ii) that we give a heuristic that seems to capture a major reason for this phenomenon.
>
> > * Algorithm 1 is DP-SGD with a regularizer, but in practice it is somewhat rare to use explicit regularization with DP-SGD.
>
> The regularizer can always be set to 0 or incorporated into the loss function, and this is how DP-SGD is usually presented for simplicity. So the regularizer does not change the validity of our heuristic.
>
> The reason we included the regularizer is that it is convenient for the counterexamples in Section 4 to not have gradient clipping. But if we instead removed clipping of the loss gradients, then the standard DP-SGD analysis would be invalid. Having an unclipped regularizer doesn’t invalidate the standard DP-SGD analysis, but allows us to include counterexamples with arbitrary gradients.
>
> > * If one of the advantages of the heuristic is that it’s less computationally expensive than privacy auditing, but you’ll have to do the computationally expensive privacy auditing anyway (in order to compare), then what is the point?
>
> One use case would be selecting hyperparameters (noise multiplier, batch size, learning rate, etc.). I.e., use our heuristic to consider many settings of the hyperparameters and select the one that is predicted to work best in terms of privacy auditing. But only actually perform privacy auditing once on the final choice of hyperparameters.
>
> > * One of the disadvantages of privacy auditing is that it’s difficult to perform correctly, but the heuristic doesn’t have rigorous theoretical guarantees. If privacy auditing and the heuristic don’t agree, then how would you know which one was wrong?
>
> That is a great question!
> The standard DP-SGD analysis gives an upper bound on $\varepsilon$, while privacy auditing gives a lower bound. In practice, there is usually a large gap between these numbers.
> The question that we face is which of the two is closer to the truth. Our heuristic gives a third number which falls between the upper and lower bounds and which has some principled justification. The heuristic thus hopefully helps answer that question.
>
> > 2. In Figure 3, why does the heuristic $\epsilon$ scale with the number of iterations $T$? Since the heuristic is a last-iterate analysis, I would have imagined that it would converge to a constant?
>
> Note that the heuristic $\varepsilon$ is minimized around $T=1/q$. At this point there is approximately a $e^{-1} \approx 0.36$ probability that the canary example is *never* sampled, which provides a lot of privacy. As $T$ increases, the probability that the canary is sampled at least once increases, but eventually it does converge.
>
> > 3. Would it be possible to extend Theorem 1 to non-regularized DP-SGD?
>
> The regularizer can be set to 0.
>
> > It’s not obvious to me why the privacy guarantee for an arbitrary loss would be similar to the privacy guarantee for a linear loss (and not even a GLM loss, mind you).
>
> This is the main thesis of our paper and it is indeed surprising!
> The main justification is our experimental results which show that linear losses do seem to represent the worst case for real deep neural networks It has been observed in the practice that the worst case examples for privacy auditing have gradients that are constant across iterations, which corresponds precisely to linear loss functions.
>
> Unfortunately, we cannot formally prove that linear losses are representative of general losses, since our counterexamples show that this is not true for pathological examples. However, as discussed in Appendix B, in the full batch setting, we can actually prove that linear losses are the worst case.
>
> > I feel like the proof of Theorem 1 is hard to appreciate without knowing more technical preliminaries
>
> We will endeavor to make the proof more accessible to different audiences.
>
> We hope that the reviewer reconsiders their recommendation in light of our responses.

---

> > ### Comment · Reviewer_8UgB · 2024-08-13
> >
> > Thanks for addressing my points. In hindsight my concerns about the regularizer don’t hold any water, but I do still have some remaining concerns.
> >
> > Using the heuristic for hyperparameter selection is an interesting idea. But I’m a little unclear on the exact set-up, and why the hyperparameters would be chosen according to privacy auditing (instead of according to utility). I imagine a scenario where for a fixed level of privacy, you try to maximize the utility by selecting the best hyperparameters subject to the privacy constraint, then do privacy auditing on the final hyperparameters. In this case I don’t see why it would be necessary to do privacy auditing on any of the hyperparameter candidates apart from the final ones.
> >
> > My other question is — if I’ve understood correctly, Theorem 1 implies that the heuristic will always be smaller than the existing $(\epsilon, \delta)$-DP bound. (Informal logic: Theorem 1 is a tight upper bound for linear losses, and extending to a larger class of loss functions can only make the privacy loss larger.) But there doesn’t seem to be a strong connection between privacy auditing and the heuristic apart from the empirical observation that the heuristic is larger than the privacy auditing bound (except for specially constructed pathological cases). I hate to say that 33,000 GPU hours aren’t enough to convince me on this, but I do feel that I would need to see the empirical observation validated across a wider range of problems and datasets (i.e., not just CIFAR10) and ideally at least some theoretical justification before I’d be comfortable agreeing that the heuristic is a valid and reliablbe tool.
> >
> > I realize that it’s not ideal to bring this up so soon before the end of the reviewer-author discussion period…but if time permits it would be great to receive clarification on these issues.

---

> ### Author Response · Authors · 2024-08-14
>
> Thank you for the comment. We respond below. We hope that we have been able to clarify all the issues before the discussion ends.
>
> > Using the heuristic for hyperparameter selection is an interesting idea. But I’m a little unclear on the exact set-up, and why the hyperparameters would be chosen according to privacy auditing (instead of according to utility).
>
> Hyperparameters would need to be chosen to achieve both high utility and low privacy loss.
> The "ideal" is to calibrate the hyperparameters so that the standard analysis of DP-SGD gives a good privacy parameter $\varepsilon$.
>
> For better or worse, it is becoming accepted practice to set the standard $\varepsilon$ to large-ish values (e.g. $10$ or $20$). One way to justify this practice is to perform privacy auditing on the final model to argue that this large $\varepsilon$ is overly conservative. This is one of the practical motivations for work on privacy auditing.
>
> The use case for hyperparameter tuning that we envisage is that one has dual privacy constraints, say, $\varepsilon_{\text{standard}} \le 10$ but also $\varepsilon_{\text{auditing}} \le 1$. Our heuristic would help evaluate the latter constraint during the hyperparameter selection phase.
> This use case is analogous to scaling laws on the utility side.
>
> That said, we reiterate that we think this work is more about basic science than about immediate applications. Progress on both sides -- provable DP upper bounds for ML and empirical privacy auditing lower bounds -- has been slowing down without converging. The goal of our work is to offer a fresh (if somewhat unorthodox) perspective -- our heuristic is neither a provable upper bound nor a lower bound; it should be somewhere in between.
>
> > if I’ve understood correctly, Theorem 1 implies that the heuristic will always be smaller than the existing
> -DP bound.
>
> Yes. This is true by the postprocessing property of DP. (The standard analysis assumes all intermediate iterates are revealed. Our result is tight for when only the sum is revealed, which is a postprocessing.)
>
> >  But there doesn’t seem to be a strong connection between privacy auditing and the heuristic apart from the empirical observation that the heuristic is larger than the privacy auditing bound
>
> This is supported by the experimental evidence, plus the observation repeated in the literature that the best privacy auditing results (i.e. worst case for privacy) are when the canary examples have the same gradient in each iteration, which corresponds to linear losses. (I.e., prior work shows that "gradient space attacks" are better than "input space attacks.") The best theoretical justification is the fact that we know that linear losses achieve the worst case in the special case of full batch training (as discussed in Appendix B).
>
> >  I hate to say that 33,000 GPU hours aren’t enough to convince me on this, but I do feel that I would need to see the empirical observation validated across a wider range of problems and datasets (i.e., not just CIFAR10)
>
> That is a fair point. But this limitation is also true of the privacy auditing literature more generally. The vast majority of experimentation is on CIFAR10 and smaller datasets. This is simply because privacy auditing generally requires many training runs, which is prohibitive for anything larger.
>
> It's important to note that the claim that our heuristic upper bounds privacy auditing results for "realistic" models & datasets is falsifiable (at least to the extent that "realistic" is clearly-defined). Our hope is that any future privacy auditing work would compare to our heuristic. If they fail to exceed our heuristic, then the claim gains more weight. If they do exceed our heuristic, then that would represent significant progress on privacy auditing.

---

### Official Review · Reviewer_6muC · 2024-07-13

**Soundness:** 3
**Presentation:** 2
**Contribution:** 2
**Rating:** 5
**Confidence:** 2

**Summary:**

The paper proposes a heuristic privacy analysis for DP-SGD that focuses on releasing only the last iterate, as opposed to all intermediate iterates. The authors argue that this approach is more realistic and provides sharper privacy guarantees in practical scenarios. The heuristic is based on a linear structure assumption for the model and is validated experimentally through attacks/privacy auditing.

**Strengths:**

This paper is well-written. The paper introduces a new heuristic analysis of DP-SGD for linear loss functions and also critically examines its limitations, and identifies areas for further research.

**Weaknesses:**

To my understanding, this paper offers a tighter privacy accounting analysis specifically for linear loss functions. However, I find its applicability limited since it cannot be extended to general ML tasks where the loss functions are not linear. Additionally, the fact that the privacy adversary has access to all intermediate iterates of the training process makes DP-SGD overly conservative is quite well-known. The main challenge remains in developing tight privacy accounting analyses for iterative algorithms like SGD.

**Questions:**

Why does the example provided in Section 4.2 show linear loss is a good approximation of many convex losses? Could you please provide some more insights?

**Limitations:**

The limitations are adequately addressed.

---

> ### Author Rebuttal · Authors · 2024-08-06
>
> We thank the reviewer for their time and comments. We address their points below.
>
> >  the fact that the privacy adversary has access to all intermediate iterates of the training process makes DP-SGD overly conservative is quite well-known.
>
> The reviewer’s claim that this phenomenon is well-known is debatable. While there are a handful of papers that prove separations between the standard analysis and the last iterate setting, there are also papers that tell the opposite story by proving that DP-SGD (with the standard analysis) is optimal in some sense. We believe that many practitioners are not aware of this limitation of the standard analysis of DP-SGD.
>
> In any case, the contribution of our work is to not only identify the phenomenon, but also to provide a heuristic that helps explain why this phenomenon arises and to study it experimentally.
>
> > I find its applicability limited since it cannot be extended to general ML tasks where the loss functions are not linear.
>
> Our manuscript is forthcoming about the fact that our analysis is only a provable bound for linear loss functions and is only a heuristic when applied to general ML tasks. Our thesis is that, in terms of privacy of DP-SGD, linear loss functions are a reasonable model for most general ML tasks, which is supported by experimental evidence.
>
> We hope that our work inspires further research into the last iterate setting of DP-SGD.
>
> > Why does the example provided in Section 4.2 show linear loss is a good approximation of many convex losses? Could you please provide some more insights?
>
> Section 4 probes the limitations of our heuristic. In Section 4.2 we tried to find a convex loss function that violates our heuristic.The largest violation we could construct is an $\varepsilon$ that is 3% higher than our heuristic. While this technically violates our heuristic, the violation is quite small, which overall supports our thesis that the heuristic is a good approximation.

---

> > ### Comment · Reviewer_6muC · 2024-08-14
> >
> > Thank you for your response. My score remains unchanged.

---

### Official Review · Reviewer_zPtg · 2024-07-18

**Soundness:** 2
**Presentation:** 2
**Contribution:** 2
**Rating:** 5
**Confidence:** 3

**Summary:**

The authors provide exact DP guarantees for cases where only the last iterate of DP-SGD is shared with the malicious clients, and linear models with linear loss functions are used. They propose their DP bound to be used as a heuristic that approximates the true DP guarantees for cases where more complex models are used. They show that for normal DP-SGD training, the predictions of their heuristic fall between the standard DP bound computed under the assumption that all intermediate iterations of DP-SGD are shared with the attacker, which is a strict upper bound of the true DP guarantee when only the last iterate is shared and DP-SGD with full batches and only last iterate sharing. They also compare their method against SoTA DP attacks and show that under most circumstances, their heuristic value for the DP is higher. They suggest that this is the result of the attacks not being good enough at precisely estimating the true DP guarantees. Finally, the authors demonstrate that their heuristic under unrealistic circumstances can underestimate the true DP guarantee but argue this only happens under hand-crafted losses and gradient updates, which do not happen in practical circumstances.

**Strengths:**

- The last iterate setting is important
- The linear function DP bound is exact
- The linear function DP bound has interesting properties
- The counter-examples for the DP heuristic themselves seem interesting and probably can be adapted to other settings

**Weaknesses:**

- I am confused by L234. The authors propose to maximize their heuristic over all $t\leq T$, while beforehand (e.g. in Figure 1/Section 2) they advocated to computing the heuristic for a single $T$. Which one is the exact proposed heuristic by the paper?
- In Figure 2, I am not sure how we adapt existing techniques to the last-iterate-only setting? Can the authors explain in more details?
- Can the authors explain in Figure 1, what network and dataset were used?
- The authors do not provide code. I am not sure about the reason, but I will give them the benefit of the doubt that the reason is indeed related to anonymity

**Nits:**
- Eq. 8. I assume you do indexing from i = 1. In that case, $A_{T-i}$ should be $A_{T-i+1}$ instead. If you do 0-based indexing, even more fixes to the equation are needed.
- I believe Eq. 7 should be multiplied by $\eta$ on the right-hand side
- Equation at L442, left-hand side should be $m_T$ not $m_t$
- I believe the last equation at L459 should have $(1-q)^{n-k}$ instead of $(1-q)^{k}$. I also believe $n$ is $T$ in this equation
- The definition of $l(m)$ in L109 is confusing as $m$ is considered input to the function, while in the rest of the paper $m$ is used as a parameter. Consider putting $x$ instead.
- Consider defining the hockey-stick divergence in terms of both its pdf and cdf in the appendix to ease unfamiliar readers. I had to read quite a bit on my own to understand it.
- Consider adding some information in the appendix as to how to deal with the mixed discrete-continuous probability for $P$. I assume many readers will be unfamiliar.
- Consider deriving the formulas for $P$ and $Q$ in Section 4.2 in the appendix. They are not obvious.
- Consider having an appendix section that quickly recaps how [NSTPC21] and [NHSBTJCT23] work. Their operation is critical for understanding Section 3. I ended up reading them to get an idea of what was going on there.

**Questions:**

(See Weaknesses Above)


While I am familiar with DP and DP guarantees, I am not a DP expert. Still, I was able to follow the math in the paper, and I am mostly confident in it. What I am not as confident in is if the heuristic, which by authors' own admission does not provide rigorous upper or lower bounds, will be useful in practice. In addition, there is few experimental results that require explanation (see above). In particular, I am concerned about why, in the counter-examples in Section 4, the authors propose maximizing the heuristic across $T$, while in the experiments in Figure 1, the heuristic is proposed to be used for a particular $T$. I am, thus, not sure which of the two versions of the heuristic should be used in practical settings as the true heuristic. Further, I am not sure if the structure of the proposed DP counterexamples in Section 4 is novel, but if it is, I think it might have uses outside of this paper.

**Limitations:**

The authors acknowledge the limitations of using the heuristic to compute the DP bounds

---

> ### Author Rebuttal · Authors · 2024-08-06
>
> We thank the reviewer for their thorough reading of our paper and thoughtful comments. In particular, we appreciate the reviewer spotting several typos, which we will fix. We respond to the main questions:
>
> > * I am confused by L234. The authors propose to maximize their heuristic over all $t \le T$, while beforehand (e.g. in Figure 1/Section 2) they advocated to computing the heuristic for a single $T$. Which one is the exact proposed heuristic by the paper?
>
> This is in Section 4, where we probe the limitations of our proposed heuristic.
> Section 4.1 illustrates the limitations of computing the heuristic for only the total number of iterations $T$. Thus in Section 4.2 we maximize over $t \le T$ to show limitations that go above and beyond what is in Section 4.1.
>
> The heuristic that we propose is simply to run with a single number of iterations $T$, but with the caveat that it is a heuristic and has known limitations, including non-monotonicity in $T$.
>
> > * In Figure 2, I am not sure how we adapt existing techniques to the last-iterate-only setting? Can the authors explain in more details?
>
> We are not sure we understand this question. The line labeled “standard” in our figures is *not* adapted to the last iterate setting. The figures illustrate that there is a large gap between the existing upper and lower bounds, which motivates our work, and our heuristic partially explains this gap.
>
> > * Can the authors explain in Figure 1, what network and dataset were used?
>
> Figure 1 does not include NN experiments (unlike the other figures). It simply compares our heuristic to some mathematical baselines in different parameter regimes.
>
> > I am not sure if the structure of the proposed DP counterexamples in Section 4 is novel,
>
> To the best of our knowledge these counterexamples are novel. However, after the NeurIPS submission deadline we have learned of subsequent related work that may reproduce some of these.
>
> > What I am not as confident in is if the heuristic, which by authors' own admission does not provide rigorous upper or lower bounds, will be useful in practice.
>
> This is a very important question. :-)
> We believe that our heuristic can be useful in practice as a “sanity check.” E.g., in practice, the standard $\varepsilon$ DP parameter is often uncomfortably large (e.g., $\varepsilon=20$) and additional validation, such as privacy auditing, is used to justify tolerating this. Our heuristic can provide additional validation, which is particularly useful because it is easy to do privacy auditing incorrectly, which gives a false sense of security.
>
> More importantly, beyond immediate practical impact, we hope that our work leads to further scientific exploration of the privacy implications of the last iterate setting. In other words, our work identifies a phenomenon and proposes an explanation (and also challenges our own explanation). Thus we believe that our work is valuable as basic science, even if the heuristic is not immediately useful.
>
> We hope that we have addressed the reviewer’s comments and that they reconsider their recommendation.

---

> ### Comment · Reviewer_zPtg · 2024-08-08
>
> **Re:** I am confused by L234.
> **Ans:** My confusion stems from the fact that by maximizing $\epsilon$, the authors take the most sound and least tight (I am using these terms here loosely, referring to their meanings for the empirical and provable DP guarantees cases, respectively) prediction they make across all $t$. Thus, despite the heuristic producing too low $\epsilon$s for many $t$, the authors report that for the “best” $t$ they are close to the counterexamples they find.
>
> **Re:** In Figure 2, I am not sure how we adapt existing techniques to the last-iterate-only setting? Can the authors explain in more details?
> **Ans:** My question is regarding the Empirical attack in Figure 2. The authors say they adapt the empirical attack to the last-iterate DP setting. My question was if the authors could provide details on how this is done?
>
>
> **Re:** Can the authors explain in Figure 1, what network and dataset were used?
> **Ans:** Can the authors describe how the plot and the heuristics themselves are compiled in more detail?
>
> **Re:** I am not sure if the structure of the proposed DP counterexamples in Section 4 is novel.
> **Ans:** I would also like to ask other reviewers who are more familiar with SOTA DP research to confirm this. Also, I would request that if they also find it novel, to take this into account in their reviews.
>
> **Re:** What I am not as confident in is if the heuristic, which by authors' own admission does not provide rigorous upper or lower bounds, will be useful in practice.
> **Ans:** I find the first part of the author’s answer here very useful. I would suggest that they include it more prominently in the paper. Regarding the second part of the answer --- I think the proposed counterexamples in Section 4 strongly suggest that general last-iterate DP guarantees are likely the same as full DP. That said, they also suggest that they are the same only for very artificial examples that only happen for extremely unrealistic models. Do the authors have any suggestions on how the DP definitions in the last-iteration case can be amended to make their worst case closer to DP-SGD executed on real networks? Maybe something related to progressing the loss during training?

---

> > ### Author Response · Authors · 2024-08-09
> >
> > Thank you for your comment. Some responses:
> >
> > > Thus, despite the heuristic producing too low $\epsilon$s for many $t$, the authors report that for the “best” $t$ they are close to the counterexamples they find.
> >
> > That is a fair point. We presented the results in Section 4.2 to show that we are capturing something more than Section 4.1, but we should also compare to the original baseline. This is what Figure 5b looks like if we just use a single $T$: [https://i.sstatic.net/IYnSlZwW.png](https://ibb.co/DpC6crS)
> > The non-monotonicity issue identified in Section 4.1 has a larger effect than the phenomenon captured in Section 4.2. (This is already evident from Figure 1c.)
> >
> > > My question is regarding the Empirical attack in Figure 2. The authors say they adapt the empirical attack to the last-iterate DP setting. My question was if the authors could provide details on how this is done?
> >
> > The prior literature on privacy auditing considers different types of attacks some of which apply to our last iterate setting and some of which don't. We don't significantly adapt the attacks, we just consider those that are applicable to the last iterate setting.
> >
> > The basic idea for all the membership inference attacks is to run the training procedure both with and without a given example and then to apply a distinguishing test to the final model.
> > For input space attacks (Figure 4), we use the loss of the malicious input as a distinguisher. For gradient space attacks (Figures 2 and 3) the distinguishing test measures the dot product of the final model checkpoint and the gradient canary.
> > The true positive rate / false positive rate tradeoff curve for this distinguishing test is then used to calculate the $\varepsilon$ values.
> >
> > The difference between Figures 2 and 3 is that in Figure 3 the gradients of the other examples are zeroed out, while in Figure 2 the other examples' gradients are nonzero and effectively act as additional noise that aids privacy. The reason we included Figure 2 is that in this setting, if the adversary has access to intermediate iterates, then prior auditing work has shown that the standard DP-SGD analysis is tight. But it is obviously very far from tight in the last iterate setting and our heuristic only partially explains the gap.
> >
> > > Can the authors describe how the plot [Figure 1] and the heuristics themselves are compiled in more detail?
> >
> > The line labelled "Heuristic" is calculated using the method described in Appendix A.1.
> > The line marked "Standard" is calculated using an open source DP accounting library [[Goo20](https://github.com/google/differential-privacy/tree/main/python/dp_accounting)].
> > The "Full Batch" line can be calculated using the same open source library or as a special case of our heuristic (we opted for the latter, since the open source library can be slow).
> >
> > > That said, they also suggest that they are the same only for very artificial examples that only happen for extremely unrealistic models. Do the authors have any suggestions on how the DP definitions in the last-iteration case can be amended to make their worst case closer to DP-SGD executed on real networks? Maybe something related to progressing the loss during training?
> >
> > This is the big question and it's something that we are continuing to work on.
> >
> > The only prior work in this direction [[FMTT18](https://arxiv.org/abs/1808.06651); [CYS21](https://arxiv.org/abs/2102.05855); [YS22](https://arxiv.org/abs/2203.05363); [AT22](https://arxiv.org/abs/2205.13710); [BSA24](https://arxiv.org/abs/2403.00278)] assumes contractivity -- i.e., if you perturb the model weights at some point, then, in subsequent steps, the effect of this perturbation does not grow or even shrinks. We can prove contractivity for smooth and (strongly) convex loss functions. But, for deep learning, we have seen experimentally that this contractivity assumption is simply not true.
> >
> > We speculate that it would be fruitful to understand to what extent real deep learning losses are "locally approximately linear" (however you want to precisely define that) and then to exploit that property in the privacy analysis. The value of our submission is that it tell us what is the best we can hope for from such an approach.

---

> > > ### Comment · Reviewer_zPtg · 2024-08-10
> > >
> > > Thank you for your comments. I would urge the authors to include the additional info in their next paper iteration.
> > >
> > >
> > > Regarding Figure 1, given the authors' comment about Appendix A1, am I correct the experiment is applied on "linear networks"?

---

> > > > ### Author Response · Authors · 2024-08-12
> > > >
> > > > Thank you for the comment.
> > > >
> > > > > I would urge the authors to include the additional info in their next paper iteration.
> > > >
> > > > We will clarify the points that the reviewers have pointed out as insufficiently clear.
> > > >
> > > > > Regarding Figure 1, given the authors' comment about Appendix A1, am I correct the experiment is applied on "linear networks"
> > > >
> > > > Figure 1 can be viewed as comparing the privacy of linear losses ("Heuristic"), general losses ("Standard"), and full batch training ("Full Batch" which is the same for linear losses and general losses) under the correspondence given by Equation 4. We don't refer to it as an experiment as it uses exact analytic formulae.
> > > >
> > > > We hope we have addressed all of the reviewer's questions and concerns about our paper.

---

> > > > > ### Comment · Reviewer_zPtg · 2024-08-12
> > > > >
> > > > > I have updated my grade in accordance with the author's comments

---

### Decision · Program_Chairs · 2024-09-25

**Decision:**

Reject

**Comment:**

The paper presents a heuristic approach for evaluating the privacy of DP-SGD when only the last model iteration is released. While the reviewers saw merit to the paper, no reviewer stood up to be a proponent of the paper. It is therefore out recommendation that the authors continue working on the heuristic and see if they could provide any rigorous bounds to their approach which should merit publication in a different venue.